# Jordan products of quantum channels and their compatibility

Mark Girard[1], Martin Plávala [2,3] & Jamie Sikora[1,4,5]

Given two quantum channels, we examine the task of determining whether they are compatible—meaning that one can perform both channels simultaneously but, in the future, choose exactly one channel whose output is desired (while forfeiting the output of the other channel). Here, we present several results concerning this task. First, we show it is equivalent to the quantum state marginal problem, i.e., every quantum state marginal problem can be recast as the compatibility of two channels, and vice versa. Second, we show that compatible measure-and-prepare channels (i.e., entanglement-breaking channels) do not necessarily have a measure-and-prepare compatibilizing channel. Third, we extend the notion of the Jordan product of matrices to quantum channels and present sufficient conditions for channel compatibility. These Jordan products and their generalizations might be of independent interest. Last, we formulate the different notions of compatibility as semidefinite programs and numerically test when families of partially dephasing-depolarizing channels are compatible.

[1] Institute for Quantum Computing, University of Waterloo, Waterloo, ON, Canada. [2] Naturwissenschaftlich-Technische Fakultät Universität Siegen, Siegen, Germany. [3] Mathematical Institute, Slovak Academy of Sciences, Bratislava, Slovakia. [4] Virginia Polytechnic Institute and State University, Blacksburg, VA, USA. [5] Perimeter Institute for Theoretical Physics, Waterloo, ON, Canada. ✉email: martin.plavala@uni-siegen.de

There are several different settings for the compatibility of states, measurements, and channels that are considered in this paper. The interested reader is referred to the reviews in refs. [1–4] for further discussions on these topics.

The quantum state marginal problem [5–8] is one of the most fundamental problems in quantum theory and quantum chemistry. One version of this problem is the following problem. Given a collection of systems $\mathcal{X}_1, \ldots, \mathcal{X}_n$ and a collection of density operators $\rho_1, \ldots, \rho_m$—each acting on some respective subset of subsystems $S_1, \ldots, S_m \subseteq \mathcal{X}_1 \otimes \cdots \otimes \mathcal{X}_n$—determine whether there exists a state $\rho$ on $\mathcal{X}_1 \otimes \cdots \otimes \mathcal{X}_n$ which is consistent with every density operator $\rho_1, \ldots, \rho_m$. For example, if $\rho_1$ acts on $S_1$, then $\rho$ must satisfy

$$\mathrm{Tr}_{\mathcal{X}_1 \otimes \cdots \otimes \mathcal{X}_n \setminus S_1}(\rho) = \rho_1, \tag{1}$$

and similarly for the other states $\rho_2, \ldots, \rho_m$. This problem is nontrivial if the density operators act on overlapping systems and indeed is computationally expensive to determine, as the problem is known to be QMA-complete [9,10]. Small instances of the problem can be solved (to some level of numerical precision), however, by solving the following semidefinite programming feasibility problem:

$$\begin{aligned} \text{find}: \quad & \rho \in \mathrm{Pos}(\mathcal{X}_1 \otimes \cdots \otimes \mathcal{X}_n) \\ \text{satisfying}: \quad & \mathrm{Tr}_{\mathcal{X}_1 \otimes \cdots \otimes \mathcal{X}_n \setminus S_1}(\rho) = \rho_1 \\ & \quad\quad\quad \vdots \\ & \mathrm{Tr}_{\mathcal{X}_1 \otimes \cdots \otimes \mathcal{X}_n \setminus S_m}(\rho) = \rho_m. \end{aligned} \tag{2}$$

Note that the condition that $\rho$ has unit trace is already enforced by the constraints. In the case where the systems $\mathcal{X}_2 = \cdots = \mathcal{X}_n$ and density operators $\sigma := \rho_2 = \cdots = \rho_n$ are identical (where we omit $\rho_1$ for indexing convenience), we remark that, for the choice of subsystems $S_i = \mathcal{X}_1 \otimes \mathcal{X}_i$ for each $i \in \{2, \ldots, n\}$, one obtains the so-called $(n-1)$-symmetric-extendibility condition for $\sigma$. This is closely related to separability testing [11].

There is an analogous task for quantum measurements called the measurement compatibility problem (see [12,13] for POVMs (positive operator-valued measures) and [14,15] for the special case of qubits). This task can be stated as follows. Two POVMs $\{M_1, \ldots, M_m\}$ and $\{N_1, \ldots, N_n\}$ are said to be compatible if there exists a choice of POVM $\{P_{i,j} : i \in \{1, \ldots, m\}, j \in \{1, \ldots, n\}\}$ that satisfies

$$M_i = \sum_{j=1}^n P_{i,j} \quad \text{and} \quad N_j = \sum_{i=1}^m P_{i,j} \tag{3}$$

for each index $i$ and $j$. In other words, the two measurements are a course-graining of the compatibilizing measurement $\{P_{ij}\}$. It may seem at first glance that both measurements are being performed simultaneously, but this view is incorrect. Performing a compatibilizing measurement should be viewed as performing a separate measurement which captures the probabilities of both measurements simultaneously. Although measurement compatibility is defined mathematically, it also has operational applications—for example, incompatible measurements are necessary for quantum steering [16–18] and Bell nonlocality [19–21].

Determining the compatibility of the two POVMs above can be solved via the following semidefinite programming feasibility problem:

$$\begin{aligned} \text{find}: \quad & P_{i,j} \in \mathrm{Pos}(\mathcal{X}), \text{ for } i \in \{1, \ldots, m\}, j \in \{1, \ldots, n\} \\ \text{satisfying}: \quad & M_i = \sum_{j=1}^n P_{i,j}, \text{ for each } i \in \{1, \ldots, m\} \\ & N_j = \sum_{i=1}^m P_{i,j}, \text{ for each } j \in \{1, \ldots, n\}, \end{aligned} \tag{4}$$

where the POVMs each act on some system $\mathcal{X}$. Note that the condition $\sum_{i=1}^m \sum_{j=1}^n P_{i,j} = \mathbb{1}_{\mathcal{X}}$ is enforced by the constraints and thus every collection of operators satisfying the conditions in Eq. (4) necessarily composes a POVM. The semidefinite programming formulation provided above may also be used to certify incompatibility by using notions of duality, i.e., to provide an incompatibility witness [22,23].

This notion of compatibility for measurements generalizes the concept for commuting measurements. Indeed, if $[M_i, N_j] = 0$ holds for each choice of indices $i$ and $j$ then the measurement operators $P_{i,j}$ defined as $P_{i,j} = M_i N_j$ form a compatibilizing POVM. This does not hold generally, as the operators $M_i N_j$ need not even be Hermitian if $M_i$ and $N_j$ do not commute. Nevertheless, one can study Hermitian versions of these matrices using Jordan products, as is discussed below.

The Jordan product of two square operators $A$ and $B$ is defined as

$$A \odot B = \frac{1}{2}(AB + BA). \tag{5}$$

This is Hermitian whenever both $A$ and $B$ are Hermitian. The Jordan product can be used to study the compatibility of measurements (see [24,25]). In particular, for POVMs $\{M_1, \ldots, M_m\}$ and $\{N_1, \ldots, N_n\}$, note that the operators defined as

$$P_{i,j} := M_i \odot N_j \tag{6}$$

satisfy

$$M_i = \sum_{j=1}^n P_{i,j} \quad \text{and} \quad N_j = \sum_{i=1}^m P_{i,j} \tag{7}$$

for each choice of indices $i$ and $j$. Thus, if $M_i \odot N_j$ is positive semidefinite for each $i$ and $j$ then the POVM defined in Eq. (6) is a compatibilizing measurement. It is known that if either $\{M_1, \ldots, M_m\}$ or $\{N_1, \ldots, N_n\}$ are projection-valued measures (PVMs), i.e., if either $M_i^2 = M_i$ for all $i \in \{1, \ldots, m\}$ or $N_j^2 = N_j$ for all $j \in \{1, \ldots, n\}$, then they are compatible if and only if each of the Jordan product operators $M_i \odot N_j$ is positive semidefinite [26]. It is also known that for POVMs, positive semidefinitiveness of $M_i \odot N_j$ is not sufficient for their compatibility [24].

Before discussing the quantum channel marginal problem, we take a slight detour and discuss the no-broadcasting theorem. A quantum broadcaster for a quantum state $\sigma \in \mathrm{Pos}(\mathcal{X} \otimes \mathcal{Y})$ is a channel that acts on the $\mathcal{X}$ subsystem of $\sigma$ and outputs a state $\rho \in \mathrm{Pos}(\mathcal{X}_1 \otimes \mathcal{X}_2 \otimes \mathcal{Y})$ that satisfies

$$\mathrm{Tr}_{\mathcal{X}_1}(\rho) = \sigma \quad \text{and} \quad \mathrm{Tr}_{\mathcal{X}_2}(\rho) = \sigma, \tag{8}$$

where $\mathcal{X} = \mathcal{X}_1 = \mathcal{X}_2$. In other words, one applies the channel that broadcasts $\sigma$ and decides afterwards where $\sigma$ is to be localized. One can show—when there is no promise on the input $\sigma$—that such a channel cannot exist due to the existence of non-commuting quantum states [27]. An easy way to see this is by trying to broadcast half of an EPR state, which would violate monogamy of entanglement. This proves the well-known no-broadcasting theorem, which states that a perfect broadcasting channel cannot exist. (The question of determining the best "noisy" broadcasting channel has also been investigated [28–30]). One may notice that the conditions above imposed by broadcasting is a special case of the quantum state marginal problem (for fixed input $\sigma$) and, moreover, of the symmetric-extendibility problem as described above (for which it is sometimes the case that no solution exists).

The task of determining compatibility of quantum channels, which has been studied recently (see, e.g. [31–35]), can be stated as follows. Given two quantum channels, $\Phi_1$ from $\mathcal{X}$ to $\mathcal{Y}_1$ and $\Phi_2$ from $\mathcal{X}$ to $\mathcal{Y}_2$, one determines if there exists another channel $\Phi$

from $\mathcal{X}$ to $\mathcal{Y}_1 \otimes \mathcal{Y}_2$ that satisfies

$$\Phi_1(X) = \mathrm{Tr}_{\mathcal{Y}_2}(\Phi(X)) \quad \text{and} \quad \Phi_2(X) = \mathrm{Tr}_{\mathcal{Y}_1}(\Phi(X)) \quad (9)$$

for every input $X$. The notion of broadcasting (as defined in the previous paragraph) is a specific instance of this problem, where one chooses both $\Phi_1$ and $\Phi_2$ to be the identity channel. One may express channel compatibility as a convex feasibility problem over the space of linear maps:

$$\begin{aligned} \text{find}: \quad & \Phi \text{ completely positive} \\ \text{satisfying}: \quad & \Phi_1 = \mathrm{Tr}_{\mathcal{Y}_2} \circ \Phi \\ & \Phi_2 = \mathrm{Tr}_{\mathcal{Y}_1} \circ \Phi. \end{aligned} \quad (10)$$

The condition that $\Phi$ is trace-preserving follows from the constraints. The channel compatibility problem (in either the no-broadcasting formulation or in the general form given in Eq. (10)) is an essential component of research in quantum cryptography. For example, if two parties, say Alice and Bob, decide to communicate via some channel $\Phi_1$, an eavesdropper Eve may use any channel $\Phi_2$ compatible with $\Phi_1$ to obtain partial information about the communication between Alice and Bob.

To see how the channel and state versions of the marginal problem are generalizations of each other, we may consider the Choi representations of the channels. It is not hard to see [32,36] that the conditions for the compatibility of $\Phi_1$ and $\Phi_2$ is equivalent to the following conditions on the Choi matrices:

$$\mathrm{Tr}_{\mathcal{Y}_2}(J(\Phi)) = J(\Phi_1) \quad \text{and} \quad \mathrm{Tr}_{\mathcal{Y}_1}(J(\Phi)) = J(\Phi_2), \quad (11)$$

where $J(\Phi_1)$ and $J(\Phi_2)$ are the Choi representations of these channels. In other words, the channels are compatible if and only if the normalized Choi matrices are compatible as states.

To see how the channel compatibility problem is a generalization of the measurement compatibility problem, consider the following reduction. For the POVMs $\{M_1, \ldots, M_m\}$ and $\{N_1, \ldots, N_n\}$, define the channels

$$\Phi_M(X) = \sum_{i=1}^{m} \langle M_i, X \rangle E_{i,i} \quad \text{and} \quad \Phi_N(X) = \sum_{j=1}^{n} \langle N_j, X \rangle E_{j,j} \quad (12)$$

where $E_{i,i}$ is the density matrix of the $i$th computational basis state; one may also think of $E_{i,i}$ as $|i\rangle\langle i|$ in Dirac notation. Channels of this form are known as measure-and-prepare (or, equivalently, as entanglement-breaking) channels. It is easy to see that the POVMs $\{M_1, \ldots, M_m\}$ and $\{N_1, \ldots, N_n\}$ are compatible if and only if $\Phi_M$ and $\Phi_N$ are compatible as channels.

The result above also holds in more general settings. In particular, a similar result holds for measure-and-prepare channels in the case when the choice of state preparations are distinguishable. However, if the preparations are chosen in a general way, then this equivalence breaks down. To be precise, consider the channels defined as

$$\Psi_M(X) = \sum_{i=1}^{m} \langle M_i, X \rangle \rho_i \quad \text{and} \quad \Psi_N(X) = \sum_{j=1}^{n} \langle N_j, X \rangle \sigma_j \quad (13)$$

for some density operators $\rho_1, \ldots, \rho_n$ and $\sigma_1, \ldots, \sigma_m$. One can show that $\Psi_M$ and $\Psi_N$ are compatible if the POVMs $\{M_1, \ldots, M_m\}$ and $\{N_1, \ldots, N_n\}$ are compatible (but the converse may not be true).

Another notion of channel compatibility that we consider in this paper concerns the case when $\Phi_1 = \Phi_2 = \Phi$ for some fixed channel $\Phi$ (i.e., when the channels are the same). A channel $\Phi$ that is compatible with itself is said to be self-compatible.

One may also consider—for some other positive integer $k > 2$—whether some choice of $k$ channels $\Phi_1, \ldots, \Phi_k$ are compatible. If $k$ copies of some fixed channel are compatible—i.e., $\Phi_1 = \cdots = \Phi_k = \Phi$ are all equal to some fixed channel $\Phi$—then we say that $\Phi$ is $k$-self-compatible.

In this work, we prove several results about channel compatibility. First, we show that the channel compatibility problem is equivalent to the quantum state marginal problem, i.e., every quantum state marginal problem can be recast as the compatibility of two channels, and vice versa. Second, we show that compatible measure-and-prepare channels (i.e., entanglement-breaking channels) do not necessarily have a measure-and-prepare compatibilizing channel. Third, we extend the notion of the Jordan product of matrices to quantum channels and present sufficient conditions for channel compatibility. These Jordan products and their generalizations might be of independent interest. Lastly, we formulate the different notions of compatibility as semidefinite programs and numerically test when families of partially dephasing-depolarizing channels are compatible.

## Results

We examine the quantum channel marginal/compatibility problem using several different perspectives. Here we provide an overview of our results, which are stated in an informal manner. Precise definitions and theorem statements can be found in Supplementary information.

**Channel compatibility generalizes the state marginal problem**. As is remarked above, the compatibility of channels is equivalent to the compatibility of their normalized Choi representations as quantum states. That is, the problem of determining the compatibility of channels can be reduced to solving the state marginal problem for a certain choice of states. We show that the quantum channel marginal problem also generalizes the state marginal problem. Recently, a somewhat weaker version of this result was proved in [36], where it was shown that the marginal problem for quantum states with invertible marginals is equivalent to the compatibility of channels. By using a different method to prove the result, we bypass the need for invertible marginals.

*Result* 1 (Informal, see Supplementary Note 2 for a formal statement.) Every quantum state marginal problem is equivalent to the compatibility of a particular choice of quantum channels.

**On the compatibility of measure-and-prepare**. We first note that every measure-and-prepare (i.e., entanglement-breaking) channel is self-compatible. Indeed, for measure-and-prepare channel $\Phi$ there exists a POVM $\{M_1, \ldots, M_m\}$ and a collection of density matrices $\rho_1, \ldots, \rho_n$ such that

$$\Phi(X) = \sum_{i=1}^{m} \langle X, M_i \rangle \rho_i, \quad (14)$$

for all $X$. Now consider the channel

$$\Psi(X) = \sum_{i=1}^{m} \langle X, M_i \rangle \rho_i \otimes \rho_i. \quad (15)$$

This channel clearly compatibilizes two copies of $\Phi$. (Moreover, one can easily modify $\Psi$ above such that it compatibilizes $k$ copies of $\Phi$. Hence every measure-and-prepare channel is also $k$-self-compatible for all $k$.)

The task of determining whether two distinct measure-and-prepare channels $\Phi_1$ and $\Phi_2$ are compatible, however, is not so straightforward. From the discussion in the previous paragraph, if $\Phi_1$ and $\Phi_2$ are expressed as measure-and-prepare channels such that the prepared states are distinct computational basis states, it can be seen that the notion of channel compatibility is equivalent to that of measurement compatibility. However, this is not the case for all measure-and-prepare channels. For instance, there may be multiple ways to express the measurements and/or preparations for a particular measure-and-prepare channel. One

might guess that any channel which compatibilizes two measure-and-prepare channels must also be measure-and-prepare and can only exist if all three sets of measurements satisfy some conditions. However, we show that this is not the case, as described below. Note that similar result was also recently obtained in a different context in [37].

*Result* 2 (See Supplementary Note 3 for details.) There exists a pair of compatible measure-and-prepare channels with no measure-and-prepare compatibilizer.

The two channels were found using semidefinite programming formulations of channel compatibility and the notion of positive partial transpose (PPT).

**Jordan products of quantum channels.** The Jordan product of two channels, $\Phi_1$ from system $\mathcal{X}$ to system $\mathcal{Y}_1$ and $\Phi_2$ from $\mathcal{X}$ to $\mathcal{Y}_2$, is the linear map $\Phi_1 \odot \Phi_2$ from the system $\mathcal{X}$ to the system $\mathcal{Y}_1 \otimes \mathcal{Y}_2$ whose Choi representation is given by

$$J(\Phi_1 \odot \Phi_2) = \sum_{i,j,k,\ell=1}^{\dim(\mathcal{X})} (E_{i,j} \odot E_{k,\ell}) \otimes \Phi_1(E_{i,j}) \otimes \Phi_2(E_{k,\ell}), \quad (16)$$

where $E_{i,j}$ is the matrix whose $(i,j)$-entry is 1 and has zeros elsewhere, i.e., one can also think of $E_{i,j}$ as $|i\rangle\langle j|$ in Dirac notation, and $\odot$ on the right-hand side denotes the Jordan product of matrices as defined in Eq. (5). It is straightforward to see that the map $\Phi = \Phi_1 \odot \Phi_2$ satisfies

$$\Phi_1(X) = \mathrm{Tr}_{\mathcal{Y}_2}(\Phi(X)) \quad \text{and} \quad \Phi_2(X) = \mathrm{Tr}_{\mathcal{Y}_1}(\Phi(X)) \quad (17)$$

for every choice of $X$. The map $\Phi_1 \odot \Phi_2$ might not be completely positive, as the corresponding Choi representation in Eq. (16) might not be positive semidefinite. However, if $\Phi_1 \odot \Phi_2$ is completely positive—and thus a channel, since it is trace-preserving by Eq. (17)—then this linear map is a compatibilizing channel for the channels $\Phi_1$ and $\Phi_2$. The condition that $\Phi_1 \odot \Phi_2$ be completely positive is therefore a sufficient condition for the channels $\Phi_1$ and $\Phi_2$ to be compatible. If $\Phi_1 \odot \Phi_2$ is not completely positive, one can use the lowest eigenvalue of the Choi matrix of $\Phi_1 \odot \Phi_2$ to estimate the amount of noise needed to add to the channels $\Phi_1$ and $\Phi_2$ to make them compatible. This amount of noise is closely related to the resource theory of compatibility and to the robustness of incompatibility of quantum channels [38]. One may also use the equivalence of the channel compatibility and quantum marginal problems from Result 1 to apply the Jordan product to the quantum marginal problem.

It is known that for projection-valued measures (PVMs)—that is POVMs where every operator is a projection—the Jordan product provides a necessary and sufficient condition for the compatibility of a PVM with any POVM [26]. It is therefore natural to consider whether the Jordan product of channels similarly provides necessary and sufficient conditions for a PVM-measurement channel to be compatible with another arbitrary channel. Namely, for a PVM $\{\Pi_1, \ldots, \Pi_m\}$, let $\Delta_\Pi$ be the corresponding measurement channel defined as

$$\Delta_\Pi(X) = \sum_{i=1}^m \mathrm{Tr}(\Pi_i X) E_{i,i} \quad (18)$$

for every choice of $X$. One may ask whether complete positivity of the Jordan product $\Delta_\Pi \odot \Phi$ is always equivalent to the compatibility of $\Delta_\Pi$ and $\Phi$, for any other choice of channel $\Phi$. This is indeed the case, as described below.

*Result* 3 (Informal, see Supplementary Note 4 for a formal statement.) $\Delta_\Pi$ is compatible with $\Phi$ if and only if $\Delta_\Pi \odot \Phi$ is completely positive.

To tackle more general cases, we describe how to relax the sufficient condition that the Jordan product be completely positive. Note that the Choi representation of the Jordan product

as provided in Eq. (16) can be expressed as

$$J(\Phi_1 \odot \Phi_2) := (\mathbb{1}_{L(\mathcal{X})} \otimes \Phi_1 \otimes \Phi_2)(A_{JP}), \quad (19)$$

where $A_{JP}$ is the Choi representation of the Jordan product of two identity channels. By replacing $A_{JP}$ in Eq. (19) with another matrix $A \in \mathrm{Herm}(\mathcal{X} \otimes \mathcal{X}_1 \otimes \mathcal{X}_2)$ satisfying

$$\mathrm{Tr}_{\mathcal{X}_1}(A) = \mathrm{Tr}_{\mathcal{X}_2}(A) = \mathrm{Tr}_{\mathcal{X}_1}(A_{JP}) = \mathrm{Tr}_{\mathcal{X}_2}(A_{JP}), \quad (20)$$

one obtains another linear map—which we denote by $\Phi_1 \odot_A \Phi_2$. Each such matrix $A$ provides another potential compatibilizing channel, so long as the corresponding map $\Phi_1 \odot_A \Phi_2$ is completely positive. If there exists at least one choice of Hermitian matrix $A$ satisfying Eq. (20) such that the map $\Phi_1 \odot_A \Phi_2$ is completely positive, we say that the channels are Jordan compatible. What is surprising is that this sufficient condition is also necessary in most cases, as described in the following two results.

*Result* 4 (Informal, see Supplementary Note 4 for a formal statement.) If the channels $\Phi_1$ and $\Phi_2$ are invertible as linear maps, then they are compatible if and only if they are Jordan compatible. (Note that the inverses do not have to be quantum channels themselves.)

The requirement that both channels are invertible is not too restrictive, as indicated by the following result.

*Result* 5 (Informal, see Supplementary Note 5 for a formal statement.) The set of Jordan-compatible pairs of channels has full measure as a subset of all compatible pairs.

**Semidefinite programs for channel compatibility.** The task of determining whether two channels $\Phi_1$ and $\Phi_2$ are compatible can be formulated as the following semidefinite programming feasibility problem:

$$\begin{aligned} \text{find}: \quad & X \in \mathrm{Pos}(\mathcal{X} \otimes \mathcal{Y}_1 \otimes \mathcal{Y}_2) \\ \text{satisfying}: \quad & \mathrm{Tr}_{\mathcal{Y}_2}(X) = J(\Phi_1) \\ & \mathrm{Tr}_{\mathcal{Y}_1}(X) = J(\Phi_2). \end{aligned} \quad (21)$$

This formulation can be found by using the Choi representations of each channel and their compatibilizer (where $X$ is the Choi representation of the desired compatibilizer).

The Jordan-compatibility of two channels can be similarly determined via the following semidefinite programming feasibility problem:

$$\begin{aligned} \text{find}: \quad & A \in \mathrm{Herm}(\mathcal{X} \otimes \mathcal{X}_1 \otimes \mathcal{X}_2) \\ \text{satisfying}: \quad & \mathrm{Tr}_{\mathcal{X}_1}(A) = \mathrm{Tr}_{\mathcal{X}_1}(A_{JP}) \\ & \mathrm{Tr}_{\mathcal{X}_2}(A) = \mathrm{Tr}_{\mathcal{X}_1}(A_{JP}) \\ & (\mathbb{1}_{L(\mathcal{X})} \otimes \Phi_1 \otimes \Phi_2)(A) \in \mathrm{Pos}(\mathcal{X} \otimes \mathcal{Y}_1 \otimes \mathcal{Y}_2), \end{aligned} \quad (22)$$

where $A_{JP}$ is the matrix as determined in the discussion around Eq. (19).

The formulation in Eq. (21) has the advantage of being linear in the Choi representations of the channels—one could therefore keep them as variables and impose affine constraints on the channels. As a concrete example, one may ask whether there exist two qubit-to-qubit channels that are compatible and both unital. (We know that two identity channels do not satisfy these two conditions, but an identity channel and a completely depolarizing channel does.)

We use duality theory to show a few theorems of the alternative for the cases of compatibility and Jordan compatibility. For an example, we present one of the two versions for compatibility, below.

*Result* 6 (Informal, see Supplementary Note 6 for a formal statement.) $\Phi_1$ and $\Phi_2$ are compatible if and only if there does not

exist $Z_1$ and $Z_2$ such that

$$\mathrm{Tr}^*_{\mathcal{Y}_2}(Z_1) + \mathrm{Tr}^*_{\mathcal{Y}_1}(Z_2) \geq 0 \quad \text{and} \quad \langle Z_1, J(\Phi_1)\rangle + \langle Z_2, J(\Phi_2)\rangle < 0. \tag{23}$$

(Note that we define formally what the adjoint of the partial trace is later, but for now it can simply be viewed as a linear map.)

**Numerically testing qubit-to-qubit channels**. We test various notions of compatibility for certain classes of qubit-to-qubit channels using the semidefinite programming formulations shown above. The family of channels that we consider are the partially dephasing-depolarizing channels defined as

$$\Xi_{p,q} = (1 - p - q)\mathbb{1}_{\mathrm{L}(\mathcal{X})} + p\Delta + q\Omega \tag{24}$$

for parameters $p, q \in [0, 1]$. Here $\mathbb{1}_{\mathrm{L}(\mathcal{X})}$ is the identity channel, $\Delta$ is the completely dephasing channel, and $\Omega$ is the completely depolarizing channel, which are defined by the equations

$$\mathbb{1}_{\mathrm{L}(\mathcal{X})}(X) = X, \quad \Delta(X) = \sum_{i=1}^{\dim(\mathcal{X})} \mathrm{Tr}(E_{i,i}X)E_{i,i}, \quad \text{and} \quad \Omega(X) = \frac{\mathrm{Tr}(X)}{\dim(\mathcal{X})}\mathbb{1}_{\mathcal{X}} \tag{25}$$

holding for all operators $X$, where $\mathbb{1}_{\mathcal{X}}$ is the identity matrix.

We investigate the values $(p, q) \in [0, 1] \times [0, 1]$ for which the channel $\Xi_{p,q}$ is $k$-self-compatible for $k \in \{1, \ldots, 10\} \cup \{\infty\}$. (Recall that the condition that $\Xi_{p,q}$ is measure-and-prepare is equivalent to the condition that it is $k$-self-compatible for all $k$.)

We also examine the values of $(p, q) \in [0, 1] \times [0, 1]$ for which $\Xi_{p,q} \odot \Xi_{p,q}$ is completely positive. This region turns out to be marginally smaller than the region where $\Xi_{p,q}$ is self-compatible, thus reinforcing the need for our generalization of the Jordan product. In fact, the channel $\Xi_{p,q}$ is invertible when $p$, $q > 0$ satisfies $p + q < 1$, so self-compatibility for this channel can be examined using (generalized) Jordan products for almost all values of $p$ and $q$.

Finally, we investigate the region of values $(q_0, q_1) \in [0, 1] \times [0, 1]$ for which the pairs of channels $(\Xi_{0,q_0}, \Xi_{0,q_1}) = (\Omega_{q_0}, \Omega_{q_1})$ are compatible and when the Jordan product $\Omega_{q_0} \odot \Omega_{q_1}$ is completely positive. It turns out that the values of $(q_0, q_1)$ for which $\Omega_{q_0} \odot \Omega_{q_1}$ is completely positive contains some nontrivial instances while simultaneously missing other trivial instances of compatible pairs. This illustrates that the (standard) Jordan product provides an interesting sufficient condition for compatibility.

## Discussion

In this work, we studied the quantum channel marginal problem (i.e., the channel compatibility problem)—which is the task of determining whether two channels can be executed simultaneously, in the sense that afterwards, one can choose which channel's output to obtain. We showed how to decide this via semidefinite programming and presented several other key properties such as its equivalence to the quantum state marginal problem.

We also studied a generalization of the Jordan product to quantum channels, and in turn, generalized it further such that it captures the compatibility of invertible channels. This Jordan product may be of independent interest.

There are many open problems concerning the compatibility of channels. We briefly mention a few which we think are interesting.

One immediate open problem is the question of whether compatibility and Jordan compatibility are equivalent. We conjecture that they are equivalent based on the fact that the set of pairs of compatible channels that are not Jordan compatible must have zero measure when the output and input spaces have the same dimension. On this note, it would be interesting to see if the resolution to this conjecture depends on the dimensions of the input and output spaces.

Another interesting problem is to examine the computational complexity of determining compatibility for a given pair of channels. Since it is equivalent to the quantum state marginal problem, we suspect that there are versions of this problem which are QMA-hard (although the equivalence is a mathematical one, and may or may not translate into efficient algorithmic reductions).

Another open problem is whether one can extend this work to study the compatibility of other quantum objects, such as quantum strategies [39–41], combs [42,43], or even channels in other generalized probabilistic theories.

Lastly, there might be a relationship between channel compatibility and cryptography. For example, symmetric extendibility is closely related to quantum key distribution, since you do not want Alice to be just as correlated/entangled with Bob and she is with Eve. Since quantum channel compatibility generalizes symmetric extendibility, perhaps there is another cryptographic setting in which the notion of channel compatibility translates into (in)security.

## Data availability

Data sharing is not applicable to this article as no datasets were generated or analyzed during the current study.

## Code availability

Code used to generate Supplementary Figs. 1 and 2 is available at https://github.com/markwgirard/Channel-Compatibility.

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

## Acknowledgements

We thank John Watrous for helpful discussions and for coining the term "compatibilizer". J.S. also thanks Anurag Anshu and Daniel Gottesman for interesting discussions about the capacity of compatibilizing channels. M.P. is thankful to Teiko Heinosaari for discussing the Jordan product of channels and the compatibility of measure-and-prepare channels. M.G. is supported by the Natural Sciences and Engineering Research Council (NSERC) of Canada, the Canadian Institute for Advanced Research (CIFAR), and through funding provided to IQC by the Government of Canada. M.P. is thankful for the support by Grant VEGA 2/0142/20, by the grant of the Slovak Research and Development Agency under Contract No. APVV-16-0073, by the Deutsche Forschungsgemeinschaft (DFG, German Research Foundation - 447948357) and by the ERC (Consolidator Grant 683107/TempoQ). J.S. is supported in part by the Natural Sciences and Engineering Research Council (NSERC) of Canada. Research at Perimeter Institute is supported in part by the Government of Canada through the Department of Innovation, Science and Economic Development Canada and by the Province of Ontario through the Ministry of Colleges and Universities.

## Author contributions

M.G., M.P., and J.S. have made substantial, direct and intellectual contribution to the work, and approved it for publication.

## Funding

## Competing interests

The authors declare no competing interests.
