## [Peer Review File · Nature Communications]

Reviewers' Comments:

Reviewer #1:

Remarks to the Author:

They authors consider the task of determining whether two quantum channels are compatible. This means that they perform both channels at the same time, and later choose one of the two channels, and they generate the output of that channel.

They present several results concerning this problem, as well as they put the problem in context.

In the beginning of the paper, they list the main points. They review the quantum marginal problem, they define the Jordan product of matrices and describe the no-broadcasting theorem. They present the quantum channel compatibility problem.

The channel compatibility problem turns out to be related to the broadcasting problem and to quantum cloning. Thus, studying it we understand more about the fundamental characteristics of quantum mechanics.

The authors present a solution for the channel compatibility problem based on semidefinite programming. They show that it is equivalent to the quantum state marginal problem.

The paper considers Jordan product channels. They derive conditions for compatibility from the Jordan product.

They show that the set of pairs of compatible channels is convex. They show that almost all pairs of compatible channels are also Jordan compatible.

Finally, the paper also analyses in detail the case of qubit channels.

The paper is very well organized, very clear. Authors identified a fundamental task of outstanding importance in quantum information science and showed an exhausting analysis of the problem, and solved it. The "channel compatibility problem" considered by the paper is as important as the no-broadcasting theorem and other already known important notions in the field. I add that Martin Plavala wrote a key paper on the subject (Ref. [29]). Thus I suggest publishing the paper in Nature Communication.

Comments:

- In Eq. (8), please check that the second equation correct.

$\text{Tr}_{\{X_i\}}(\rho_1) = \sigma$

In particular, is the subscript "1" necessary. I would expect that it is not needed.

- Maybe, one could mention explicitly that the the broadcasting means that we want to broadcast a density matrix from a set of two, noncommuting matrices, e.g., in the formulation of H. Barnum, C. M. Caves, C. A. Fuchs, R. Jozsa, B. Schumacher, Phys.Rev.Lett. 76 (1996) 2818-2821.

Reviewer #2:

Remarks to the Author:

Submitted paper presents results on the compatibility of quantum channels. I find it the most comprehensive paper up to date on the topic. It includes both a basic summary of current results

and presents new ones through a newly developed construct of Jordan product of quantum channels. It is a valuable paper with many new results - equivalence of channel compatibility with the quantum state marginal problem, non-existence of measure-and-prepare compatibilizing channel for measure-and-prepare channels in general and the fact, that the positivity of Jordan product of two channels might be both necessary and sufficient condition for all pairs of channels (up to a set of zero measure).

These results are presented in a very approachable way. I highly appreciated simplified explanations prior to their presentation - many chapters had short explanations of results at their beginnings and the paper as a whole presented main results in a separate chapter 2. This improves readability at least for me. Besides this point the flow of the paper is logical and the results are sound and clearly explained. I have found no point for any concern.

The only question that remains lingering in my head that could have been discussed (there is also possibility I missed it) is the following:

- There is an equivalence between measurement compatibility and a specific subset of measure-and-prepare channels.
- There is a clear separation between compatibility of measurements and the Jordan product condition for these measurements.
- The Jordan product condition of two (randomly chosen) channels is (almost surely) also a necessary condition for the two channels.

If the last condition would be holding for measure-and-prepare channels, we would have a good condition for determining compatibility of two measurements. The reason why this is not so is evident from the paper, but a clear discussion would be beneficial, I believe.

To summarize, I find the paper of a very good quality with many new interesting results. I find no major issues anywhere in the paper and I suggest the editor to proceed with publishing the paper after the authors consider the (optional) point I raised.

Daniel Reitzner

Reviewer #3:

Remarks to the Author:

The authors introduced a task of determining whether two given quantum channels with the same input space are compatible with each other, i.e., whether we can find a third channel from the same input space to the tensor product of their respective output spaces such that when ignoring one output we can always exactly implement the other quantum channel. In other words, one channel can be used to realize both channels in a compatible way. This is somewhat straightforward generalization of the measurement compatibility problem, discussing whether two different POVMs can be obtained from a third POVM in a compatible way. In order to solve this problem, a notion of Jordan product of two quantum channels are introduced, again as a generalized of the similar notion for two linear operators. The authors then obtained a number of mathematical results, and here I just list a few major ones which I feel of some interests:

- 1) The quantum channel compatibility problem is equivalent to a special case of the quantum marginal problem, which has been well studied and is QMA-complete in general. (this result seems quite straightforward from the definition of channel compatibility but the proof still contains some tricks)
- 2) The authors exhibit a pair of compatible measure-and-prepare channels (these are equivalent to entanglement-breaking channels) which are not compatible via a measure-and-prepare way (Example 15).

3) When two channels have inverse maps, they are compatible if and only if they are Jordan-compatible. This further leads the authors to show that the set of Jordan-compatible pairs of channels has full measure as a subset of all compatible pairs.

4) This quantum channel compatibility problem can be formulated in various SDP forms (see Eq. (23) as a notable form), thus can be decided by various available SDP toolkits.

I also feel the whole paper is well written and is organized in an accessible way despite the technical nature of these results. I have not checked all technical details (especially those numerical examples) but I find their proofs are quite convincing. I also believe the notion of Jordan product for channels/operators should deserve more attention, and could be a potentially useful tool in quantum information science. However, I also see there is an obstacle that the authors have to address in a convincing way. I don't see any further operational interpretation of the channel compatibility problem in addition to the definition. The Jordan product of channels also needs further justification for its usefulness and significance. I feel the current version is more suitable for a mathematical journal rather than nature communication. The authors mentioned a few possible lines for future study. It would be nice to see some more discussions about the significance or applications of their results.

Response to referees and list of changes

We are grateful to the referees for careful reading of our manuscript and the suggestions and criticism that lead to its improvement. In the revised version, we have tried to follow all the raised points. We include bellow detailed responses to each point raised by the referees and a list of changes.

Reviewer 1

1. In Eq. (8), please check that the second equation correct. $Tr_{\Xi_2}(\rho_1) = \sigma$ In particular, is the subscript “1” necessary. I would expect that it is not needed.

Indeed, this was a typo. We thank the reviewer for spotting it.

2. Maybe, one could mention explicitly that the the broadcasting means that we want to broadcast a density matrix from a set of two, noncommuting matrices, e.g., in the formulation of H. Barnum, C. M. Caves, C. A. Fuchs, R. Jozsa, B. Schumacher, Phys.Rev.Lett. 76 (1996) 2818-2821.

Note and citation to the suggested paper was added to Introduction.

Reviewer 2

1. There is an equivalence between measurement compatibility and a specific subset of measure-and-prepare channels.

We address this equivalence in Proposition 14.

2. There is a clear separation between compatibility of measurements and the Jordan product condition for these measurements.

The separation was mentioned in the Introduction and respective paper was cited.

3. The Jordan product condition of two (randomly chosen) channels is (almost surely) also a necessary condition for the two channels. If the last condition would be holding for measure-and-prepare channels, we would have a good condition for determining compatibility of two measurements. The reason why this is not so is evident from the paper, but a clear discussion would be beneficial, I believe.

We thank the referee for this comment. We added discussion of Jordan compatibility of measurement channels after Theorem 40. Since the dimension of the output Hilbert space of measurement channel depends on

the number of elements of the given POVM, our results are not significantly applicable here. But we will consider the special case of measurement and measure-and-prepare channels in the future.

Reviewer 3

1. I don't see any further operational interpretation of the channel compatibility problem in addition to the definition.

Our aim is not to investigate additional operational interpretations of channel compatibility problem, but to show that it is equivalent to quantum marginal problem for states and provide possible solutions to the problem in form of Jordan product. We added a note that channel compatibility either in the form of no-broadcasting theorem, or in the general form is important for quantum cryptography.

2. The Jordan product of channels also need further justification for its usefulness and significance. It would be nice to see some more discussions about the significance or applications of their results.

We added a note that one can use the Jordan product of channels to estimate the robustness of incompatibility in the resource theory of channels, and that one can use the equivalence between quantum marginal problem and channel compatibility to apply Jordan product to the quantum marginal problem, so the results create a synergy that is yet to be explored in future work. Apart from that, we also show that Jordan product can be used to construct joint channels of partially depolarizing-dephasing channels in non-trivial cases.

List of changes

The following includes list of major or untracked changes. Other changes are tracked and highlighted in the manuscript.

- Symbols denoting authors affiliations were changed to numbers.
- Conclusions section was renamed Discussion and moved to appropriate place.
- Paper organization section was removed.
- Sections 4 - 10 were changed to subsections of Methods section.
- Data Availability section was added.
- Code Availability section was added.
- Subsections removed from Introduction section.
- Introduction edited to fit Nature Communications style.
- Indexes of POVMs changed so that POVM M_i is indexed by $\{1, \dots, m\}$ and N_j is indexed by $\{1, \dots, n\}$.

- The notation of zeros in matrixes was unified.
- Footnotes in Results were moved to main text.
- Subsection removed from Discussion section.
- Italics were removed from the main text.
- Styles of theorems and proofs were changed according to the editorial request.
- Bibliography style was changed according to the editorial request.